# Intelligent Online Inspection of the Paste Quality of Prebaked Carbon Anodes Using an Anomaly Detection Algorithm

**Laiyi Li, Qingzong Li, Wentao Yong, Shuwei Zhang, Maolin Yang** and **Pingyu Jiang** *

State Key Laboratory for Manufacturing Systems Engineering, Xi'an Jiaotong University, Xi'an 710049, China; yongwentao@stu.xjtu.edu.cn (W.Y.)

* Correspondence: pjiang@mail.xjtu.edu.cn

**Abstract:** Prebaked carbon anodes are a critical consumable in the aluminum electrolysis industry. Prebaked carbon anode paste is the intermediate product of the prebaked carbon anode, and its quality significantly impacts the prebaked carbon anode. Therefore, inspecting the quality of the prebaked carbon anode paste is essential. Currently, the quality inspection of the paste still relies on laboratory analysis or manual experience. A laboratory inspection cannot obtain results in real time, while manual inspection poses potential risks. To address these issues, an online intelligent inspection method for prebaked carbon anode paste based on an anomaly detection algorithm was proposed. Firstly, we acquired the temperature of the paste and the power of the kneading motor. Secondly, we transformed these time-series data into images using the Gramian Angular Field (GAF) technique and joined them to create the paste anomaly detection dataset. Thirdly, we trained a matched anomaly detection model based on the PatchCore algorithm. Finally, we compared two advanced models: HaloAE and TSRD. PatchCore performs best on our dataset with an AUC-ROC score of 0.9943, followed by HaloAE (0.9906) and TSRD (0.9811). Our proposed method enables on-time intelligent inspection of prebaked carbon anode paste quality. This eliminates the need for manual inspection, reduces labor requirements, and ensures worker safety.

**Keywords:** prebaked carbon anode paste; quality inspection; anomaly detection; Gramian Angular Field

## 1. Introduction

As a carbon product, prebaked carbon anodes are the critical consumable in the electrolysis process for aluminum production owing to their high electrical conductivity and chemical stability [1,2]. A prebaked carbon anode is a mixture of strongly heated petroleum coke and coal-tar pitch, causing the pitch to bind the coke particles together [3]. The general production process of a prebaked carbon anode includes calcining, pulverizing, screening, milling, kneading, forming, baking, etc. The kneading process plays an important role, distributing the particles uniformly to produce anode paste with appropriate plasticity, making them desirable for the forming process. Prebaked carbon anode paste is the intermediate product of a prebaked carbon anode, and the paste is composed of dry coke aggregates, fine coke particles, and pitch as a binder [4]. The paste must have good fluidity and uniformity to ensure the uniform distribution of various particles, resulting in a prebaked carbon anode with high strength and better conductivity. Therefore, assessment of prebaked carbon anode paste quality through online intelligent inspection is significant.

Currently, two main approaches are used for paste quality inspection: laboratory inspection and manual inspection. The laboratory can test the paste's mechanical properties, thermostability, and other factors. However, this approach is time-consuming, despite its accuracy in analyzing paste anomalies. In the prebaked carbon anode process, the kneaded paste needs to be promptly transferred to the next process. The laboratory inspection method does not give results until production is complete and thus is untimely

in controlling intermediate product quality. In this case, the unqualified paste will continue to be put into the following process, leading to production waste. The manual inspection method can determine the quality of paste in a timely manner. However, it heavily relies on the experience of workers. During the inspection, workers must wear gloves to inspect the freshly produced paste by hand. The paste temperature is as high as 160 °C, which causes high labor intensity and potential safety hazards. Hence, the implementation of intelligence algorithms for paste quality inspection is necessary.

Intelligent algorithms are extensively employed in product quality inspection, which can detect the quality of the paste intelligently. However, the paste exhibits a low anomaly frequency, resulting in a lack of sufficient anomaly samples. The dataset may suffer from an imbalance of positive and negative samples if a classification algorithm is used. In this case, anomaly detection techniques are employed to address this issue. Anomaly detection (AD) refers to the problem of identifying patterns in data that deviate from the expected behavior, indicating anomalies or outliers. This technique enables semi-supervised or unsupervised training of data without requiring specific attention to the adequacy of the anomalous samples. The training data amount for the AD model is typically less than 500, such as each category on the MVTec AD benchmark only consisting of around 200 to 500 data samples. As the anomalous samples are solely used to evaluate the AD model, we can acquire a substantial amount of normal paste sample data for model training and only need a small amount of normal and anomalous samples for evaluation.

The quality of the dataset plays a crucial role in the training of a high-performing model, and the basis of the dataset determines the quality of the dataset. Analyzing the kneading process of the paste, the normal paste temperature trend during the kneading process ensures the chemical properties of the paste, such as thermostability. The proper operation of the kneading motor ensures thorough mixing and contact between petroleum coke and coal-tar pitch, thus ensuring mechanical properties. Therefore, the quality of the paste can be inferred from the aforementioned factors during the kneading process. For this purpose, paste temperature data and kneading motor power data were collected as the basis of our dataset.

In this paper, an online intelligent inspection method is proposed for prebaked carbon anode paste quality based on an anomaly detection algorithm. Firstly, we acquired the temperature data of the paste and the power data of the kneading motor during manufacturing. Subsequently, the acquired data were cleaned and labeled. These time-series data were then transformed into images using Gramian Angular Field (GAF) and then joined to build a dataset for deep learning. Next, the parameters of the paste anomaly detection model based on the PatchCore algorithm were fine-tuned to achieve optimal detection performance. Then, two anomaly detection algorithms, including HaloAE and TSRD, were compared with PatchCore to determine the most effective model for inspecting paste quality anomalies. The results show that the PatchCore algorithm achieves an AUC-ROC score of 0.9943 on our dataset, followed by HaloAE (0.9906) and TSRD (0.9811). This shows that the PatchCore algorithm has better learning power for our dataset and better classification performance of the model. In the P-R curve, the PatchCore algorithm exhibits a remarkable ability to balance precision and recall. Therefore, the PatchCore algorithm demonstrated superior performance across all aspects.

In summary, the contributions of this paper are:

- The proposed method addresses the issue of online intelligent inspection for prebaked carbon anode paste. An anomaly detection method is employed to address the problem of insufficient anomalous samples in paste production, a common issue in industrial product quality inspection.
- This method provides a feasible approach to dataset construction for prebaked carbon anode paste quality control. It transforms time-series data into 2D images using the GAF technique, where each image corresponds to a pot of anode paste, enabling the application of computer vision detection algorithms for paste quality inspection.

## 2. Related Works

### 2.1. Time-Series Anomaly Detection

The paste production data acquired in this paper are time-series data, which should theoretically use time-series data anomaly detection methods.

Anomaly detection in time-series data is an important research topic. In recent years, extensive research has been conducted on anomaly detection in time-series data. Wei et al. [5] proposed a hybrid deep learning model that combines long short-term memory (LSTM) with an autoencoder (AE) for anomaly detection tasks regarding indoor air quality, in which the LSTM network learns the long-term dependencies of the data in a time-series sequence, and the AE identifies the optimal threshold based on the reconstruction loss rates evaluated on every data across all time-series sequences. Chadha et al. [6] employed a 1D convolutional neural-network-based deep autoencoder architecture, whose contribution is to split the autoencoder latent space into discriminative and reconstructive latent features, which is superior to standard autoencoder architectures. Lin et al. [7] proposed a VAE-LSTM hybrid model as an unsupervised approach for anomaly detection in time series, where VAE was utilized as a local feature extractor. Li et al. [8] proposed a MAD-GAN framework for multivariate anomaly detection, using the LSTM-RNN as the base model to capture the temporal correlation of time-series distributions. Niu et al. [9] proposed an LSTM-based VAE-GAN for time-series anomaly detection before the general GAN-based method, in which the mapping process brought errors and took a long time. TAnoGan [10] and TadGAN [11] are GAN-based anomaly detection approaches in time series that have been proposed recently. Zhan et al. [12] proposed a multiscale stacked spatial-temporal graph attention network for multivariate time-series anomaly detection (STGAT-MAD), coherently capturing the feature and temporal correlations among multivariate time-series data by stackable STGAT networks. Wang et al. [13] proposed a multiscale transformer-based residual variational autoencoder model (MT-RVAE), in which the transformer model can extract the correlation between different features.

The inputs in these methods are arrays with labels or points with labels. Due to the non-cyclical and independent nature of paste production data, each datum cannot be split up. Directly using them as array inputs will encounter the issue of varying data quantities in each production. So, it is not possible to use labeled arrays as input. On the other hand, the point input method requires labeling each point, but the time-series points do not have direct correspondence with the paste quality. As a result, it is hard to build a dataset.

In contrast, algorithms for detecting anomalies in images only require labeling the entire image rather than each pixel. The label of each image corresponds precisely to the quality of the paste, and images of various sizes can be readily standardized. Transforming data into images enables the utilization of high-performance computer vision algorithms to detect time-series data. For this purpose, the image anomaly detection method is more suitable for the paste quality online intelligent inspection method.

### 2.2. Image Anomaly Detection

Anomaly detection in computer vision has been an important research field for a long time. A lot of deep-learning-based image anomaly detection algorithms have been developed for applications in medicine and industry. Jain et al. [14] proposed a deep CNN-based model, WCENet, for image anomaly detection and localization in medical fields. Liu et al. [15] proposed an anomaly detection method for hyperspectral data named SSC-AE, which combines a self-supervised classifier and autoencoder to improve the performance of standard autoencoders, enabling the detection of quality defects in strawberries. There are also many anomaly detection algorithms developed based on the MVTec AD benchmark that have good application prospects. Mathian et al. [16] introduced a novel HaloNet-based local transformer autoencoder, denoted as HaloAE, which incorporates transformers to enhance the anomaly detection capabilities of the AE. Deng and Li [17] introduced the "reverse distillation" paradigm along with a trainable first-class bottleneck embedding module, incorporating them into the teacher–student model (TSRD). PatchCore

was introduced in 2021 as an anomaly detection technology to enable the full recall of defective products in industrial manufacturing. Roth et al. [18] optimized the nearest-neighbor-based anomaly detection method proposed by Bergman et al. [19] and Cohen and Hoshen [20]. This latest image anomaly detection method shows great advantages in the industrial cold start-up process. Ishida et al. [21] proposed SA-PatchCore to detect anomalies in co-occurrence relationships and local areas via a self-attention module based on PatchCore. In recent years, generative models have also been used in anomaly detection due to the advantage of being able to generate samples. GAN-based models have been well developed in the field of image anomaly detection. The representative and influential models include AnoGAN [22], EGBAD [23], and GANomaly [24]. The diffusion model detects anomalies by learning the diffusion pattern of noise added to the sample under normal conditions. It can achieve image sample quality superior to the current state-of-the-art generative models [25]. In the medical field, the diffusion model is often applied in image anomaly detection. AnoDDPM is a novel partial diffusion anomaly detection strategy based on denoising diffusion probabilistic models with superior mode coverage over GANs and higher sample quality than VAEs, utilizing simplex noise instead of Gaussian noise to capture larger anomalies. Wyatt et al. [26] applied the AnoDDPM to 22 tumor datasets of T1-weighted MRI scans (CCBS Edinburgh) and obtained good results for anomaly detection of brain tumors. Similarly, Wolleb et al. [27] proposed a weakly supervised anomaly detection method based on the denoising diffusion implicit model and applied it to brain CT tumor examination.

The literature reviewed above indicates that HaloAE, TSRD, and AnoDDPM are reconstruction-based methods, but sometimes a network can "generalize" so well that it can also reconstruct the anomalies, leading to misdetection. The GAN-based methods are based on synthesizing anomalous features, but these features from synthetic data tend to stray far from the normal features, resulting in poor model performance. In contrast, PatchCore is an embedding-based method that achieves state-of-the-art performance. It uses pre-trained ImageNet to extract features and learn normal feature distribution and tests anomalies by comparing the input features with the memorized features. This approach is reliable but requires matching pre-trained networks with parameters. In our method, we will adjust the type of pre-trained network and training parameters in PatchCore for our dataset.

### 2.3. Time Series to Image Transformation

The dataset for the image anomaly detection algorithm is in image format. Therefore, we need to transform the time-series data from the temporal dimension to the spatial dimension, converting it into images. With this in mind, this section provides a literature review of different methods for transforming time-series data.

The most popular techniques for encoding 1D time-series data into 2D images in recent works are the Gramian Angular Field (GAF), recurrence plots (RP), and the Markov transition field (MTF). Yang et al. [28] adopted the Gramian Angular Summation Field (GASF) and Gramian Angular Difference Field (GADF) to encode time series as images so that the deep learning architectures could be applied to learn the structures and patterns of the image. In light of this, Liu et al. [29] combined near-infrared spectroscopy with GAF image coding and successfully achieved the qualitative classification of different diesel fuels and alcohol content detection using neural networks. Lee et al. [30] transformed the acquired inertial measurement unit data into images using GAF and performed automatic detection of excessive carrying load on workers during various construction tasks. Wu et al. [31] developed a system prediction model based on GAF data transformation, which can realize contextual prediction of power systems for economic dispatch. Ma et al. [32] also applied GAF in the medical field to transform heart rate data into images for classification and detection. Li et al. [33] transformed time series into recurrence plots to form time-series images from which local features can be extracted. Xu et al. [34] proposed an unsupervised anomaly detection model based on NVAE for univariate time series, namely T2IVAE,

which transforms the 1D time series into a 2D image as input to capture more information on time-series correlation. Time-series images were generated using RP, GAF, and MTF algorithms for time-series data in some works [35–37]. Wen et al. [38] developed a time-domain signal-to-image transformation method in which raw signals fulfill the pixels of the image by sequence, known as the gray pixel image. The grayscale image method was also adopted [39–41]. A color symmetrized dot pattern (CSDP) method [42] was newly designed to infuse three multisensor signals into images, handling multisensor information fusion problems. The CSDP method was proposed based on the conventional SDP. It can infuse three different signals to form a colorful image, and each signal represents one RGB channel.

Given the strong temporal correlation of GAF images, they effectively capture the features of time-series data. Moreover, the practical application of this method has increased in recent years. Therefore, using GAF as the data transformation method for online intelligent paste quality inspection is a logical choice.

## 3. A New Method of Quality Anomaly Detection

The new method of prebaked carbon anode paste quality anomaly detection is shown in Figure 1.

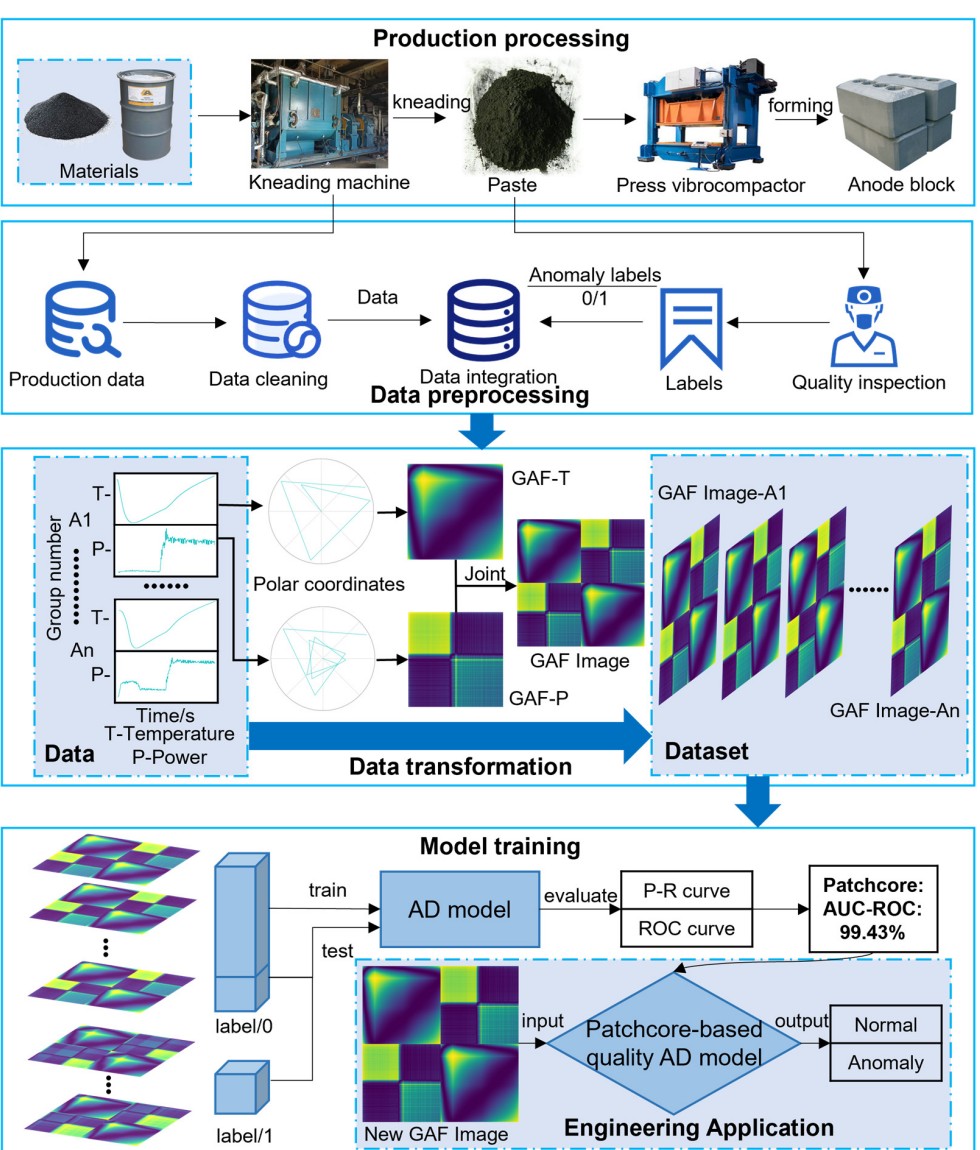

**Figure 1.** The new method of prebaked carbon anode paste quality anomaly detection.

The production process of a prebaked carbon anode involves two main stages: (1) Kneading raw materials, including petroleum coke and coal-tar pitch, in a kneading machine to create prebaked carbon anode paste. (2) Transforming the prebaked carbon anode paste into the press vibrocompactor to shape it. First of all, temperature and power sensors were installed in the kneading machine and motor, respectively, to collect production data for building our dataset. In total, 555 sets of paste samples were collected.

The second step is data preprocessing. The raw production data are first cleaned and then labeled based on the quality inspection results of the corresponding paste by the workers. The data consist of 550 sets labeled 0 and 5 sets labeled 1.

Thirdly, the temperature data and power data in 555 sets are transformed into images with pixels of $128 \times 128$ by means of GAF. The GAF images of temperature and power are then joined together into $2 \times 2$ images in the angular direction, which serve as the deep learning input for the quality anomaly detection model. The data processing and dataset details will be presented in Section 4.

Finally, the dataset is partitioned and input into the anomaly detection algorithms for training to obtain quality anomaly detection models. We compared three anomaly detection models, and the PatchCore algorithm demonstrated superior performance across all aspects for the paste quality online intelligent inspection method.

## 4. Data Processing and Dataset

### 4.1. Data

The manufacturing process of the paste is a process of fully integrating the aggregate with the binder; the laboratory inspection mainly tests the pastes' mechanical properties and thermostability and assesses their quality. The normal heating process uniformly bonds a binder to an aggregate to obtain a product with good thermal stability. Less heat will result in the binder not being sufficiently melted, while more heat will result in the binder having a lower viscosity for bonding purposes. Therefore, the temperature variation of the paste determines its thermostability. The normal operation of the kneading motor makes the friction and fusion between the aggregate and the aggregate, and between the aggregate and the binder, and finally obtains a paste with strong mechanical properties. So, the working condition of the kneading motor determines the mechanical properties of the paste, and the data that can directly reflect the working condition of the mixing and kneading motor are the power data. For this purpose, paste temperature variation data and kneading motor power data were collected as the basis of our dataset.

The temperature variation data of the paste and the power data of the kneading motor are acquired as time-series data during the kneading process. The sampling interval is set to 1.5 s, and each production process takes approximately 1–1.5 h, so each data table contains about 3000 points.

Figure 2 shows the representation of temperature and power data in Cartesian coordinates and the overall data structure. The dataset comprises 555 sets of data categorized as 0 or 1, representing qualified or unqualified paste quality, respectively. The data for each kneading process are divided into two stages by the red line in Figure 2. The first stage is known as dry kneading, in which the cold aggregates, such as petroleum coke, are first put into the pre-heated kneading machine to be kneaded. The temperatures first drop rapidly and then rise slowly after reaching the equilibrium temperature of about 100 °C. The power of the kneading motor is kept at a steady state during this phase, as the quantity and nature of the raw materials remain unchanged. The second stage is called wet kneading, in which pre-heated liquid coal-tar pitch is added to the kneading machine, and an increase in the slope of the temperature data can be noticed from the red dividing line. The sudden increase in the power data is due to the fact that the viscous coal-tar pitch requires a great deal of motor power to be fully kneaded.

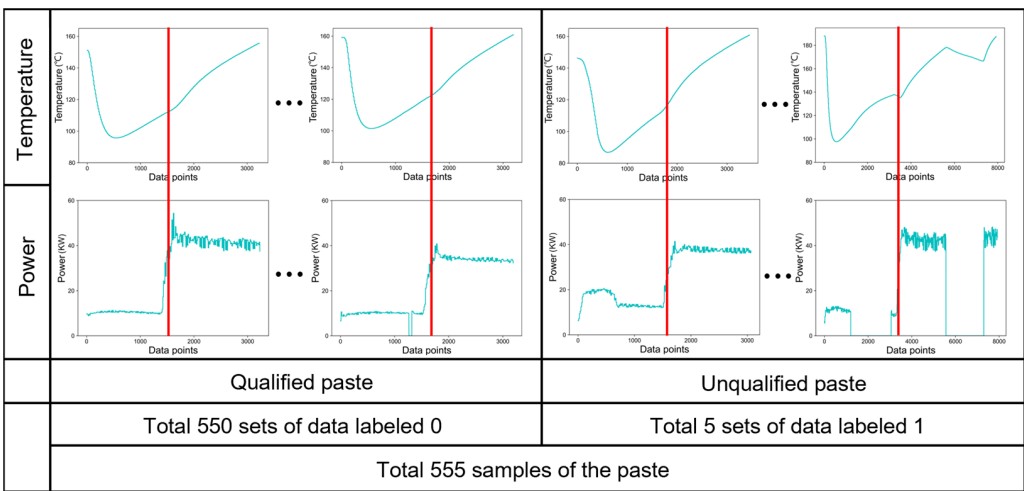

**Figure 2.** The representation and structure of the data.

### 4.2. GAF Transformation and Dataset

As a widely used feature-mapping method, GAF has the advantage that it can preserve temporal dependency. The GAF can correlate data features at the pixel level and code 1D time-series data into 2D images without losing any features. Therefore, it is possible to apply GAF to image anomaly detection. The integrated data are transformed into image data using a GAF using the following steps:

First, each temperature or power time series was normalized into the interval [0, 1]:

$$X_n^T = \left\{ x_1^T, x_2^T, \ldots, x_i^T, \ldots, x_N^T \right\}, \ X_n^P = \left\{ x_1^P, x_2^P, \ldots, x_i^P, \ldots, x_N^P \right\}, n \in [1, 555] \tag{1}$$

$$\widetilde{X} = \frac{(x_i - \min(X))}{\max(X) - \min(X)}, (i = 1, 2, 3, \ldots \ldots, N) \tag{2}$$

where $N$ is the total number of data points, $n$ is the number of integrated sets, $T : Temperature$, and $P : Power$.

Next, the rescaled time series $\widetilde{X}$ was transformed into polar coordinates by encoding the value as an angular cosine and the time stamp with the equation as follows:

$$\begin{cases} \varphi = \arccos(\widetilde{x}_i), 0 \leq \widetilde{x}_i \leq 1, \widetilde{x}_i \in \widetilde{X} \\ r = \frac{t_i}{N}, i \in [1, N] \end{cases} \tag{3}$$

where $\varphi$ is the angle value, $r$ is the radius, and $t_i$ is the time stamp.

For temperature or power data in each set of integrated data, a new operation was defined: $x_i \oplus x_j = \cos(\varphi_i + \varphi_j)$, where $\varphi_i, \varphi_j$, respectively, represent the corresponding angle of $x_i, x_j$ in the polar coordinates. The GAF matrix was finally defined:

$$\begin{aligned} GASF = [\cos(\varphi_i + \varphi_j)] &= \begin{bmatrix} \cos(\varphi_1 + \varphi_1) & \cdots & \cos(\varphi_1 + \varphi_N) \\ \vdots & \ddots & \vdots \\ \cos(\varphi_N + \varphi_1) & \cdots & \cos(\varphi_N + \varphi_N) \end{bmatrix} \\ &= \widetilde{x}_i \cdot \widetilde{x}_j - \sqrt{1 - \widetilde{x}_i}^2 \sqrt{1 - \widetilde{x}_j}^2 = \widetilde{X}^T \widetilde{X} - \sqrt{1 - \widetilde{X}^2}^T \sqrt{1 - \widetilde{X}^2} \end{aligned} \tag{4}$$

The temperature and power time-series data were all transformed into 2D images using the aforementioned GAF method in the polar coordinate system. From Equations (2) and (3), it can be observed that the transformed angle takes values in the range $[0, \pi]$, and the cosine value monotonically decreases in this range for a given time-series data. Thus, when the data are mapped to polar coordinates, only a unique mapping result is produced.

From Equation (4), it can be observed that the elements in the GAF matrix move from left to right and from top to bottom as the value of $\varphi$ increases, thus maintaining the temporal correlation.

Furthermore, the 555 sets of time-series data containing temperature and power were transformed into 1110 GAF images with $128 \times 128$ pixels. To fully utilize the multimodal data, the GAF images of the two data were joined in the diagonal direction to form a new image, which serves as the input for the deep learning model. The data processing and dataset are shown in Figure 3. Ultimately, the dataset consists of 555 images with $256 \times 256$ pixels, with 550 normal images labeled as 0 and the remaining five anomalous images labeled as 1.

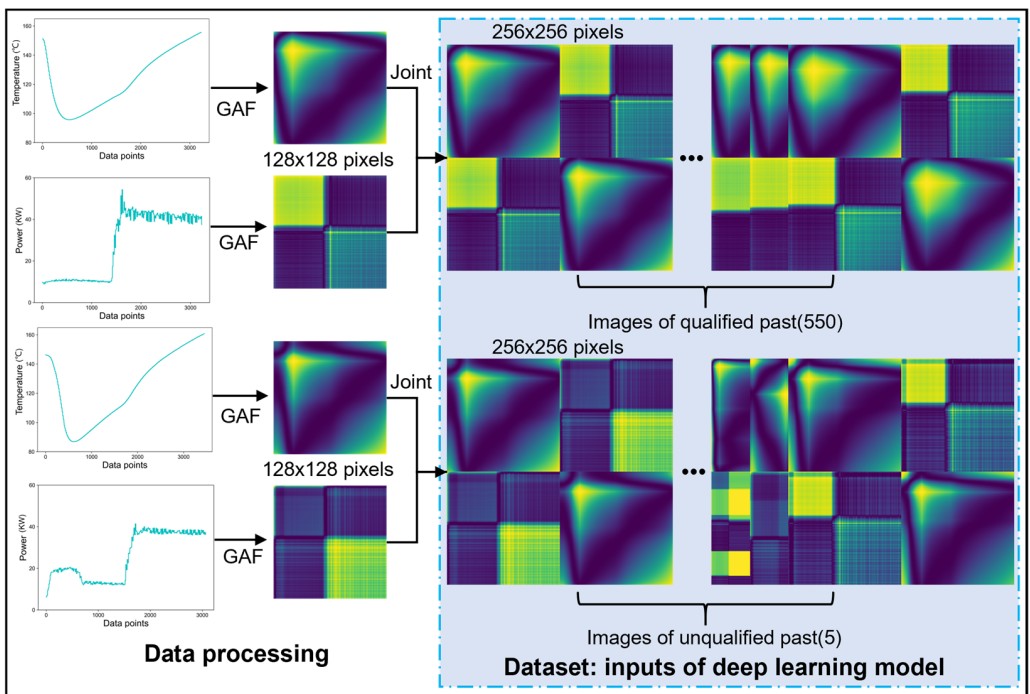

**Figure 3.** Data processing and dataset.

## 5. Result and Comparison

### 5.1. PatchCore Algorithm

PatchCore is an anomaly detection algorithm based on image feature distribution and clustering, capable of providing detailed anomaly detection results and interpretable anomaly features. Its structure consists of two main parts, as shown in Figure 4, and is described as follows:

(1) Feature patches extraction: the local perceptual features in the normal image will be extracted by the feature extractor to obtain the feature patches of all images.

(2) Core set sampling and anomaly scoring: All feature patches are fed into the K-nearest neighbor (KNN) for core feature sampling. The core feature set obtained from the sampling will form the memory bank, which is the criterion for calculating the image anomaly score. During the test, patch features are extracted for the test sample, and the anomaly scores are calculated using the KNN and the memory bank.

PatchCore utilizes deep features and core set sampling, showcasing its diverse capabilities in time-series analysis, resilience against various anomaly types, computational efficiency, interpretability, and scalability.

During the initial stage of feature patch extraction, we altered the pre-trained network type to densenet201. Then, we assessed detection performance after feature extraction with different layer configurations. Subsequently, we adjusted the parameters associated with

the sampling rate and KNN. Finally, we obtained the optimal model parameters that suit our dataset.

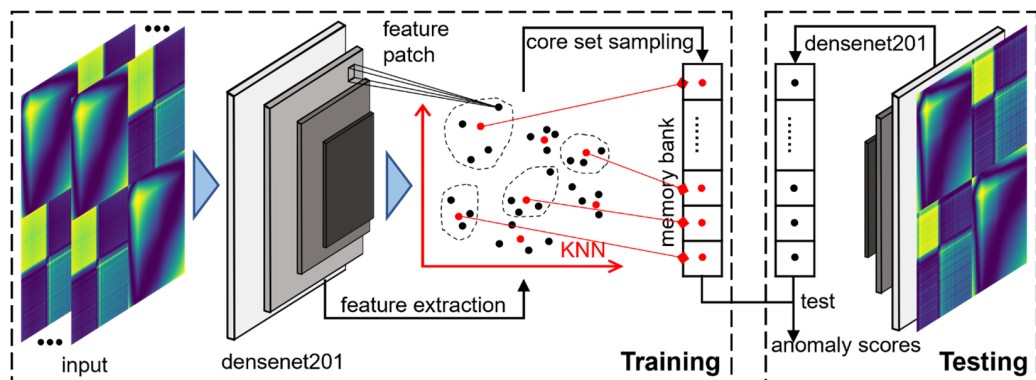

**Figure 4.** PatchCore architecture.

### 5.2. Model Training and Result

In this section, we segment the dataset and feed it into the PatchCore algorithm to train the anomaly detection model.

Dataset partitioning: The training set consists of 80% of the dataset (444 images in total), and all of their labels are 0. The testing set (111 images in total) consists of the remaining images, in which 106 images' labels are 0 and 5 images' labels are 1. Among them, the data labeled 0 are divided by random sampling to avoid the problem that the machine production process and working conditions are too similar in continuous periods. We use the t-sne (t-Distributed Stochastic Neighbor Embedding) method to display the raw data distribution in Figure 5, where we can find that the training and testing set are uniformly distributed, and the anomalous data are also doped in the distribution.

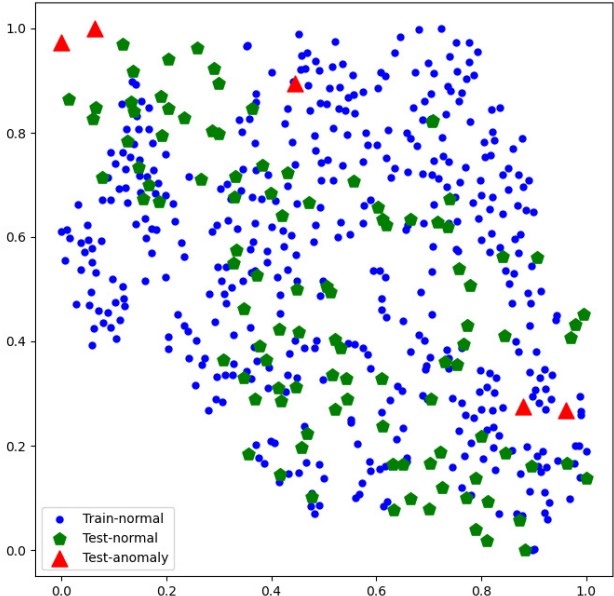

**Figure 5.** The raw data distribution.

Operating Environment: The experiment platform is built on a deep learning workstation with an Intel Xeon silver 4216 CPU (Intel Corporation, Santa Clara, State of California, the United States) and Nvidia GeForce RTX™ 3090 GPU (Nvidia Corporation, Santa Clara, State of California, the United States). The operating environment of the workstation is shown in Table 1.

**Table 1.** Operating environment.

| Project | Project Content |
|---|---|
| Operating system | Microsoft Windows 10 |
| Python environment | Python 3.8 |
| Virtual environment | Anaconda3-2022.10 |
| Programming software | PyCharm 2021.3.1 |

Model Training: In the initial stage of model training, the training set is read in RGB mode, and all input image tensor dimensions are (3,256,256). When training the model, we set the number of random seeds to 40. Each training and testing iteration requires approximately 5 min.

The experiments used three pre-trained neural networks that performed well. The identical training parameters were set as in Table 2, and the AUC-ROC scores obtained with consistent parameters are presented in Table 3. It can be found that Densenet201 performs best on our dataset.

**Table 2.** The identical training parameters.

| Feature Layers | Core Set Percentage | Nei. Agg. Size | Neighbors |
|---|---|---|---|
| 2 and 3 | 10% | 5 | 3 |

**Table 3.** The AUC-ROC scores obtained with consistent parameters.

| Network | Wideresnet50 | Resnext101 | Densenet201 |
|---|---|---|---|
| AUC-ROC | 0.9811 | 0.9830 | 0.9849 |

Subsequently, the model was trained, and its parameters were fine-tuned utilizing the Densenet201 network. Table 4 illustrates the parameter adjustments made during the model training by comparing the AUC-ROC scores to the optimal parameters. The data in Table 4 indicate that the optimal AUC-ROC score of 0.9943 was achieved with feature layers 2 and 3, a neighborhood aggregation size of 3, a neighbor number of 1, and a 10% core set percentage.

**Table 4.** The parameters during the model training.

| Network | Feature Layers | Core Set Percentage 10% (Nei. Agg. Size, Neighb., AUC-ROC) | Core Set Percentage 1% (Nei. Agg. Size, Neighb., AUC-ROC) |
|---|---|---|---|
| densenet201 | 2 and 3 | (3, 1, 0.9943) (5, 3, 0.9849) | (3, 1, 0.9906) (5, 3, 0.9868) |
| | 3 and 4 | (3, 1, 0.9774) (5, 3, 0.9887) | (3, 1, 0.9887) (5, 3, 0.9925) |

## 5.3. Comparison

In this section, we have selected the HaloAE and the teacher–student model in reverse distillation (TSRD) for comparative analysis. Finally, our parameter-tuned PatchCore model has demonstrated superior performance across all aspects.

Five metrics were used to evaluate the classification ability of the models: accuracy (*Acc*), precision (*Pr*), sensitivity (*Se*) or true positive rate (*TPR*) or *Recall*, specificity (*Sp*), and F1 score (*F1*). Finally, the models were evaluated using the Precision–Recall (P-R) curve and the Receiver Operating Characteristic (ROC) curve. The P-R curve can assess the model's ability to balance accuracy and recall rates, which is important in industrial anomaly detection. The ROC curve can evaluate the anomalous detection ability of models. The ROC score was obtained by analyzing the test results to derive the true positive rate

(*TPR*) and false positive rate (*FPR*), with the area under the curve (AUC) computed in the ROC graph. The equations of the metrics are as follows:

$$Acc = \frac{(TN + TP)}{(TN + TP + FN + FP)} \tag{5}$$

$$Pr = \frac{TP}{(TP + FP)} \tag{6}$$

$$Se/TPR/Recall = \frac{TP}{TP + FN} \tag{7}$$

$$Sp = \frac{TN}{(TN + FP)} \tag{8}$$

$$F1 = \frac{2 \times Se \times Pr}{(Se + Pr)} \tag{9}$$

$$FPR = 1 - Sp = \frac{FP}{TN + FP} \tag{10}$$

where *TP* refers to true positive, *FP* refers to false positive, *TN* refers to true negative, and *FN* refers to false negative.

The confusion matrices of these models are depicted in Figure 6. By examining the main diagonal of these matrices, a detailed comparison of Figure 6a–c reveals that the PatchCore model exhibits the lowest inspection error rate among the three methods. Table 5 lists the classification performance metrics for each model. It is observed that all three models—PatchCore, HaloAE, and TSRD—achieve 100% recall for anomalous samples when tested with the optimal threshold. Even with a small test sample, PatchCore achieves an F1 score of 0.9091 and an accuracy of 0.9910. The HaloAE and the TSRD perform lower than PatchCore in each of the remaining metrics.

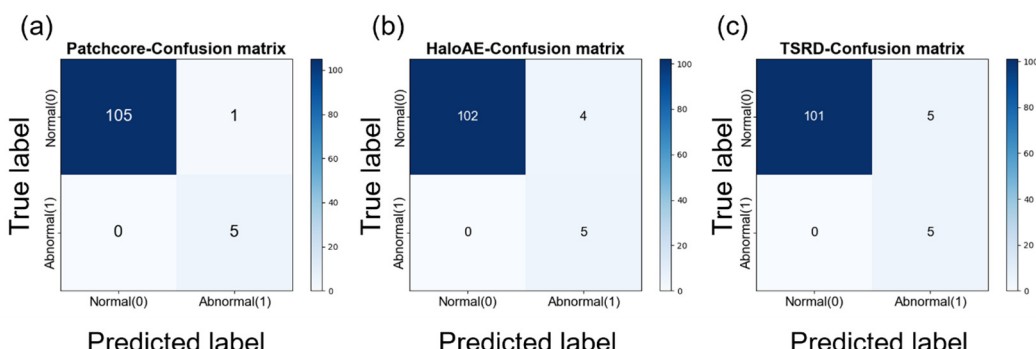

**Figure 6.** The confusion matrices of (**a**) PatchCore, (**b**) HaloAE, and (**c**) TSRD.

**Table 5.** The metrics for the classification performance of three models.

| Models | Accuracy | Precision | Recall | Specificity | F1 Score |
|---|---|---|---|---|---|
| PatchCore | 99.10% | 83.33% | 100.0% | 99.10% | 90.91% |
| HaloAE | 96.40% | 55.56% | 100.0% | 96.23% | 71.43% |
| TSRD | 95.50% | 50.00% | 100.0% | 95.28% | 66.67% |

Figures 7–9 represent the Precision–Recall (P-R) curves for the anomaly of the three models. By comparing these figures, it can be observed that when the recall is equal to 1.0, the detection precision of PatchCore is 0.8333, which is higher than that of the HaloaAE and TSRD by about 0.5556 and 0.5000, respectively. This indicates that PatchCore achieves both high recall and high precision in anomaly detection on our dataset.

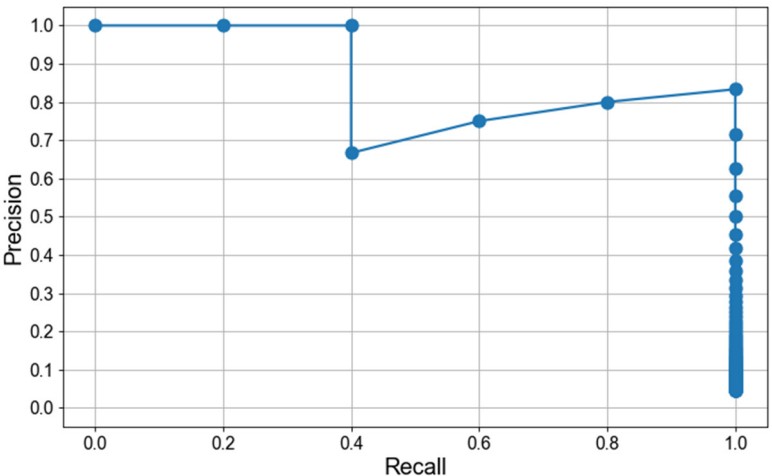

**Figure 7.** The Precision–Recall curve for the anomaly of PatchCore.

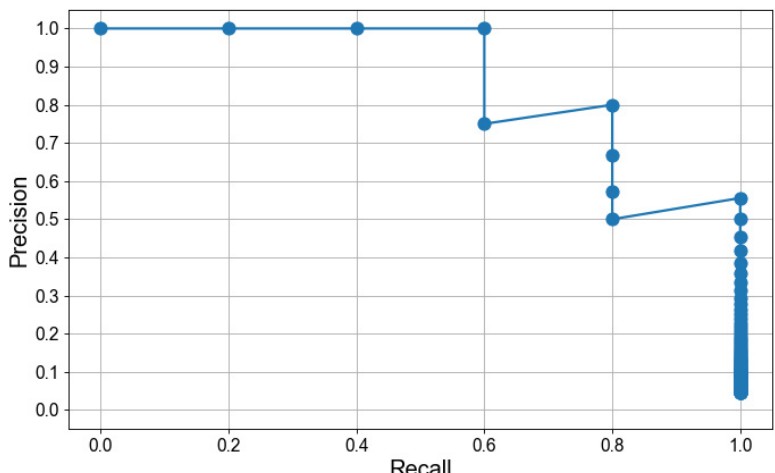

**Figure 8.** The Precision–Recall curve for the anomaly of the HaloAE.

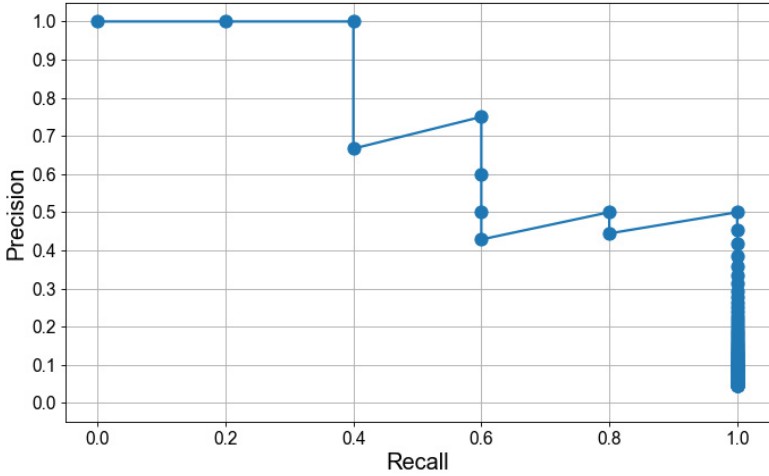

**Figure 9.** The Precision–Recall curve for the anomaly of the TSRD.

Figure 10 displays the AUC-ROC curves for the three models, and the area under the curve represents the ROC score. The ROC score of the PatchCore model is 0.9943, whereas the HaloAE and TSRD have ROC scores of 0.9906 and 0.9811, respectively. The larger area

under the AUC-ROC curve proves the better classification performance of the model. As a special note, multiple threshold points need to be taken when plotting this curve.

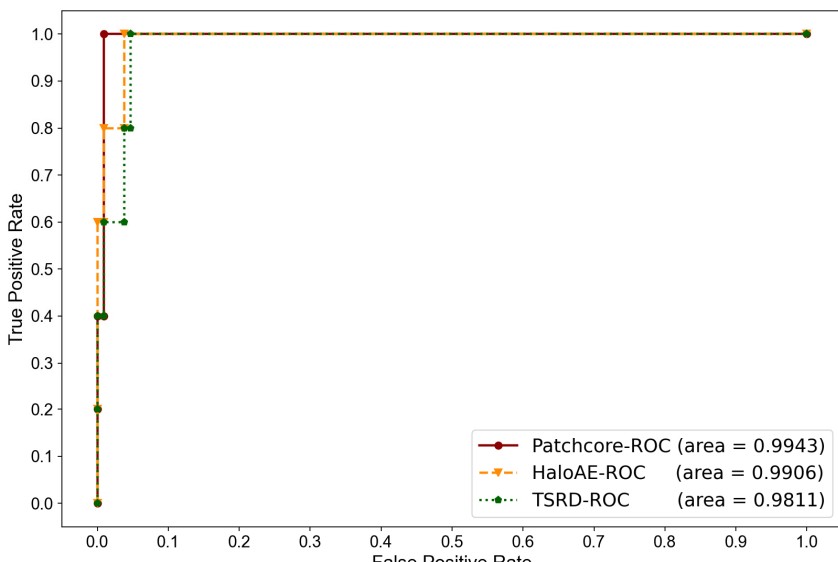

**Figure 10.** The AUC-ROC curves of PatchCore (area = 0.9943), HaloAE (area = 0.9906), and TSRD (area = 0.9811).

Figure 11 shows the descriptive statistics and normality tests of the three models for the 111 test samples. The statistical criteria are the anomaly scores of the sample, which were normalized. The blue marginal rugs in the figure represent normal samples, and the red marginal rugs in the red box are anomaly samples. It can be observed that the Gaussian distributions fitted to the samples in Figure 11a are more concentrated, and there is less overlap between normal and abnormal samples compared to Figure 11b,c. Therefore, PatchCore has a better classification performance than the other two models, and its higher AUC-ROC scores are due to the fact that there are fewer normal samples distributed among the anomaly class. Additionally, the normal samples of the three models conform to a Gaussian distribution, proving that the features of our dataset are well learned by the models. The normal samples of PatchCore have a more concentrated Gaussian distribution, which proves that PatchCore has better learning ability for our data than the other two models. Figure 12 shows the t-sne reduction of the data in the PatchCore model. There were 1600 features per sample at the time of testing, and we performed feature reduction and eventually embedded the samples in a 2D space. It can be noticed that the normal and anomaly samples are well separated, and the same as in Figure 11a, only one normal sample is doped into the anomaly class.

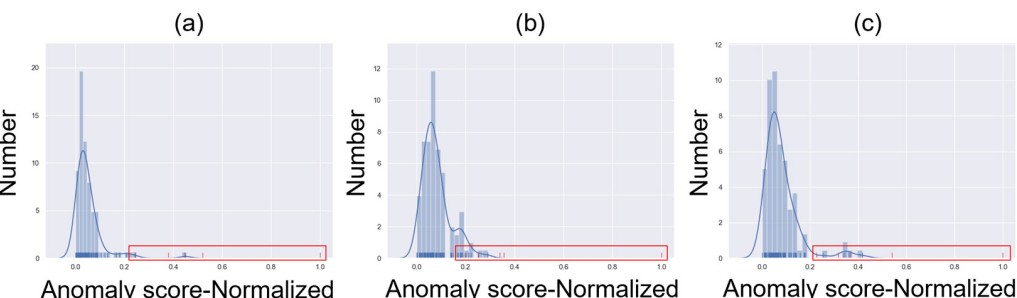

**Figure 11.** Descriptive statistics and normality tests for test set. (**a**) PatchCore, (**b**) HaloAE, (**c**) TSRD. The blue marginal rugs in the figure represent normal samples, and the red marginal rugs in the red box are anomalies.

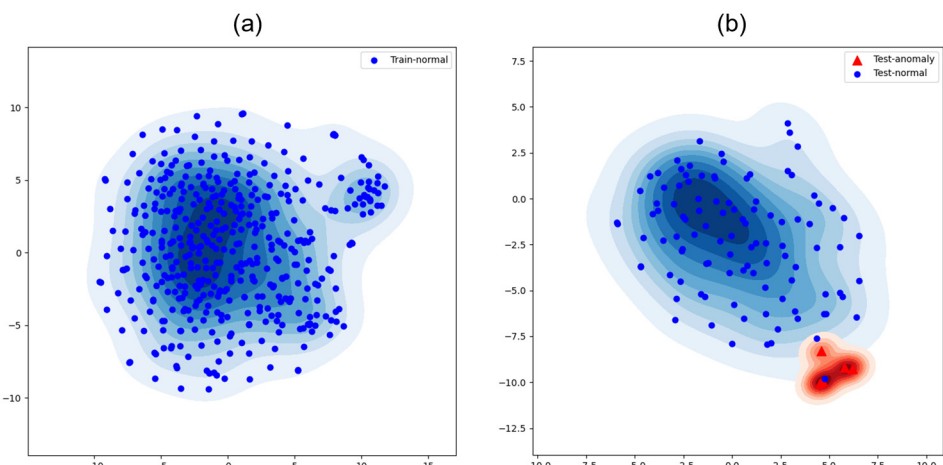

**Figure 12.** The t-sne reduction of the data in the PatchCore model. (**a**) T-sne for Training set, (**b**) T-sne for Testing set.

These results indicate that all three anomaly detection models perform well, and the PatchCore model has demonstrated superior performance across all aspects. Therefore, our proposed method for inspecting paste quality anomalies can effectively detect paste quality by using the GAF to transform time-series data into images.

In our endeavor to evaluate and compare the performance of our trained PatchCore model, we initially compared several classification metrics to evaluate the classification performance. Subsequently, we compared the P-R curves and AUC-ROC curves of these three models to determine anomaly detection performance. Our primary focus was to maximize recall while also improving accuracy and model generalization. The PatchCore model achieved these objectives, making it the preferred choice for the paste quality online intelligent inspection method.

## 6. Conclusions

This study presents an online intelligent inspection method for prebaked carbon anode paste quality based on an anomaly detection algorithm. Firstly, the temperature and power data were acquired for the dataset. Secondly, the production data were transformed into images by using GAF, and then the two kinds of images were joined to form the dataset before training the anomaly detection model. Thirdly, we trained the paste anomaly detection model using the PatchCore algorithm, adjusting the network type and optimizing parameters to achieve the optimal detection performance. Finally, two popular algorithms were compared with the PatchCore model. Our model, with a 0.9943 AUROC score for anomaly detection, shows better learning power for our dataset and better classification performance. In the P-R curve, the PatchCore algorithm exhibits a remarkable ability to balance precision and recall. In the final visualized test results, it is known that our new method of paste quality anomaly detection based on PatchCore is effective.

This paper contributes to solving the problem of online intelligent inspection for prebaked carbon anode paste quality. Our proposed method addresses the drawbacks of the time-consuming laboratory inspection and the experience-dependent manual inspection conducted in harsh working environments. The method employed a GAF to construct an image format dataset from the paste production data, offering a novel approach to dataset construction for prebaked carbon anode paste quality control. Additionally, we successfully address the challenge of insufficient unqualified paste samples, a common issue in industrial product quality inspection, by employing an anomaly detection algorithm. The results demonstrate that our proposed method achieves prebaked carbon anode paste quality through online intelligent inspection, which can determine paste quality in a timely manner. This eliminates the need for manual quality inspection, reduces labor requirements, and ensures worker safety.

The limitation of this paper is that it can only determine whether the quality of the paste is qualified but cannot provide direct causes for the anomalies. In the future, we will collect more paste samples and will classify them based on the categories of anomalies. Subsequently, the data features in each category of anomalies should be analyzed, and a new model can be developed using these data. This new model should accurately identify categorized anomalies and also possess the capability to detect unknown anomalies.

**Author Contributions:** Conceptualization, P.J.; data curation, W.Y.; formal analysis, L.L.; funding acquisition, P.J.; investigation, S.Z.; methodology, L.L. and Q.L.; resources, P.J.; software, L.L.; supervision, P.J.; validation, L.L. and Q.L.; writing—original draft, L.L., W.Y. and S.Z.; writing—review and editing, Q.L., M.Y. and P.J. All authors have read and agreed to the published version of the manuscript.

**Funding:** This research was supported by the National Key Research and Development Program of China under Grant No. 2021YFE0116300.

**Data Availability Statement:** The datasets generated during and analyzed during the current study are not publicly available.

**Conflicts of Interest:** The authors declare that there are no conflict of interest related to this research.

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
