# Peer review of "Intelligent Online Inspection of the Paste Quality of Prebaked Carbon Anodes Using an Anomaly Detection Algorithm"

_systems, doi:10.3390/systems11090484_

Round 1
Reviewer 1 Report
1. 1. Based on the results that the Patchcore algorithm achieves an AUC-ROC score of 0.9943 on their dataset, followed by HaloAE (0.9906) and TSRD (0.9811). In the P-R curve, the Patchcore algorithm exhibits a remarkable ability to balance Precision and Recall. The authors , therefore, draw the conclusion that the Patchcore algorithm demonstrated superior performance across all aspects. Here, it is better for authors to work out the significance an AUC-ROC score of 0.9943 compared to HaloAE (0.9906) and TSRD (0.9811), that is what is the meanings of the differences (0.0037) between the AUC-ROC score 0.9943 for and HaloAE score 0.9906? Without sound scientific evaluation of the significance of this small difference, it is difficult to convince that the arguments and discussion of findings are coherent, balanced and compelling.
2. Authors believe that their method provides a feasible approach for the quality inspection dataset in industrial production. There are no industrial application validation tests to support this claim.
3. The authors use the paste temperature and motor power as the data source for data transformation and produce the datasets for model training, it is better to add more why they should be used, not the other data. Otherwise, it is difficult to say the content is succinctly described and contextualized with respect to previous and present theoretical background and empirical research.
4. After addressing the above three concerns, it is suggested the authors modify their conclusion wordings to reflect the above.
Besides,
1. This paper is to design and develop an online intelligent inspection method for prebaked carbon anode paste based on an anomaly detection algorithm, it is better and more logic to give the title of Section 3 A New Method of Quality Anomaly Detection, not a framework. A method is different from a framework.
2. 2.2. Time Series to Image Transformation should be 2.3. Time Series to Image Transformation, wrong section number.
1) English is understoodable
2) Better to smooth English.
Reviewer 2 Report
This paper presents the use of different ML methods for detecting anomalies in paste creation. It's clear that the methods work however I have some concerns about how few anomalies are available for testing. I wonder if the authors could create more test data through linear combinations of the existing anomalies. Then see if the algorithm can detect these as also anomalous. It would also be useful to do something like a t-sne reduction of the data and see of the training and test data are overlapping or not. Finally some additional figures showing the distribution of the training data would improve the presentation quality of the paper. I also suggest cutting down some of the background that may be extraneous.
The grammar can be improved. There are a number of sentences that seem somewhat disjoint from the preceding thought or are incomplete. Please revise.
